# Long-Distance Passive Sensing Tag Design Based on Multi-Source Energy Harvesting and Reflection Amplification

**DOI:** 10.3390/mi16010018

**Published:** 2024-12-26

**Authors:** Gang Li, Chong Pan, Bo Wu, Zhiliang Xu, Shihua Li, Yehua Zhang, Yongjun Yang, Zhuohang Zou, Chang Shi, Muze Wang

**Affiliations:** 1Ji An Electronics Integrated Circuit and Communication Transmission Laboratory Technology Co., Ltd., Ji’an 343000, China; wubo1113@126.com (B.W.); xzl3322503@126.com (Z.X.); m18802063426@163.com (S.L.); yemajiayou@163.com (Y.Z.); 2School of Information and Software Engineering, University of Electronic Science and Technology of China, Chengdu 610054, China; 3State Grid Sichuan Electric Power Company, Chengdu Power Supply Company, Chengdu 610041, China; enoch19861212@126.com; 4Centre for RFIC and System, School of Information and Communication Engineering, University of Electronic Science and Technology of China, Chengdu 611731, China; zouzhuohang2023@163.com (Z.Z.); 202322011533@std.uestc.edu.cn (C.S.); 5School of Integrated Circuit Science and Engineering, University of Electronic Science and Technology of China, Chengdu 611731, China; 202321011512@std.uestc.edu.cn

**Keywords:** energy harvesting, RFID, passive sensor tag, reflection amplification, multi-source energy harvesting

## Abstract

Wireless sensor networks often rely on battery power, which incurs high costs, considerable volume, and a limited lifespan. Additionally, the communication range of existing passive sensor tags remains short, which challenges their suitability for evolving Internet of Things (IoT) applications. This paper, therefore, presents a long-distance passive RFID sensing tag that integrates multi-source energy harvesting and reflection amplification. Multi-source energy harvesting enhances tag receiving sensitivity and extends the system’s downlink communication distance, while reflection amplification increases tag reflection power and improves the uplink communication distance, thereby significantly expanding the overall communication range. The test results show that the tag achieves a receiving sensitivity of −45 dBm, a reflection gain of 44 dB, and a communication distance of up to 96 m.

## 1. Introduction

With the rapid development of Internet of Things (IoT) technology and increasing requirements for lower cost, smaller size, and minimal power consumption, passive sensing tags and RFID-based systems have grown to be popular academic alternatives.

In 2015, J. S. Lee and associates at Seoul National University, South Korea, developed a passive wireless hydrogen concentration sensing system that integrates RFID technology with a platinum/graphene composite [1]. The hydrogen sensor, placed on the RFID tag antenna, exhibits impedance variations in response to hydrogen concentration fluctuations, which in turn alters the frequency and power of the reflected signal from the tag. By monitoring the changes in these reflected signal parameters, the system can infer hydrogen concentrations. However, due to the sensor’s impedance mismatch with the tag antenna and circuitry, the effective communication range is limited to 0.25 m, posing challenges for large-scale hydrogen monitoring applications. In 2020, D. Inserra and associates at the University of Electronic Science and Technology of China proposed a screw-loosening detection technique utilizing UHF RFID tags [2]. This method enables detection of the loose (open) or tight (close) states of metal screws by observing variations in the tag’s reflection coefficient, as changes in the screw’s state affect the impedance matching between the tag’s antenna and circuitry. Nevertheless, this design limits the communication range to 1.3 m. In 2015, Z. B. Xiao from Fudan University developed a wireless immobilized sensing tag tailored for continuous glucose monitoring, integrating a glucose sensor, RFID front end, 10-bit ADC, and a temperature sensor [3]. The glucose detection range reaches 0–30 mM, with a sensitivity of 0.75 nA/mM, yet the power sensitivity remains low at −4 dBm, whose theoretical working distance is 2.6 m. In 2010, J. Yin and colleagues at the Hong Kong University of Science and Technology introduced an on-chip passive UHF RFID tag with an integrated temperature sensor, utilizing inverted packaging technology to combine the custom-designed tag chip with an antenna [4]. Though the tag exhibits limited sensitivity at −6 dBm, the sensor’s temperature detection range spans from −20 to 30 °C, with a maximum communication range of 4 m. In 2021, Z. Shao and colleagues from Shanghai Jiao Tong University introduced a passive RFID-based tag for influenza measurement, incorporating a magnetic field sensor to detect the magnetic field surrounding a cable, thereby inferring the current corresponding to magnetic field strength [5]. This tag achieves a maximum communication distance of 5.2 m. In 2011, J. Virtanen and colleagues from the Rauma Research Institute in Finland introduced a passive UHF RFID sensing system incorporating a humidity sensor fabricated via inkjet printing techniques [6]. This sensor, printed on a Kapton HN polyimide substrate, measures humidity by detecting changes in the polyimide’s permittivity, achieving a maximum communication range of 8.7 m.

The above method of relying only on RF energy harvesting for power supply limits the sensitivity of sensor tags, thus reducing the working distance of the system. An effective way to improve the sensitivity of sensor tags is to adopt multi-source energy harvesting technology, among which the combination harvesting technology of RF energy and solar energy has been extensively studied [7,8,9,10,11,12].

In 2016, A. E. Abdulhadi and colleagues at McGill University, Canada, proposed an RFID sensing tag capable of dual energy harvesting from both solar and RF sources [11]. This system consists of an RFID chip, a microcontroller unit (MCU), and temperature and humidity sensors, and it achieved a communication range of 7.48 m when powered by RF energy, extending to 27 m with solar energy. Yet, the system’s MCU is restricted to sensor data harvesting without full RFID protocol processing, thus limiting its operational flexibility. Furthermore, this study primarily focused on the performance characteristics of a single sensor tag, without providing a detailed analysis of sensor tag performance in a multi-tag configuration based on discrete components.

It is challenging to enhance the receiver sensitivity of a collector in RFID systems, regarding self-interference issues, which constrains the system’s uplink communication range. To address the limited communication distance resulting from the collector’s low receiver sensitivity, a reflection amplifier can be incorporated into the tag. This approach leverages the amplifier’s negative resistance properties to increase the tag’s reflection coefficient beyond 1, therefore effectively boosting the reflected signal power from the tag. Consequently, the reverse (uplink) communication range can be extended without modifying the collector’s receiver sensitivity [12,13,14,15,16,17,18,19,20,21].

F. Farzami et al. employed a tunnel diode to achieve a reflection amplifier with high power-added efficiency [14]. Utilizing the tunnel effect to generate negative resistance, this design enables a reflection coefficient greater than 1, thereby increasing reflected power. With an input frequency of 0.89 GHz and input power of −30 dBm, the reflection gain reached 17 dB, alongside a power-added efficiency of 24.56%, but the reflection gain remained modest. Additionally, F. Amato et al. proposed a passive RFID tag structure integrated with a tunnel diode [21], but their implementation was limited to a reflection amplifier rather than a full tag system. Under input conditions of 5.45 GHz and −70 dBm, the reflection gain achieved was 34.4 dB.

Unlike the approaches in the literature [14,21] which employ tunnel diodes to realize reflection amplifiers, J. Kimionis et al. in 2014 explored the use of a crystal transistor (BFT25A) to achieve this functionality [14] and advanced a range-enhanced RFID tag integrating solar cells, reflection amplifiers, and RFID chips. This method uses positive feedback to generate negative resistance, enabling a reflection coefficient greater than 1, thus intensifying reflection power. As a result, only the reflection amplifier was realized, with no fulfillment of the full range-enhanced RFID tag. Notably, under an input frequency of 900 MHz and input power of −50 dBm, the reflection gain reached 29 dB, with a power-added efficiency of only 1.65%, leaving room for further optimization in both reflection gain and power-added efficiency. In the same year, J. Kimionis et al. further proposed a range-enhanced RFID tag that incorporates a reflection amplifier along with a binary phase-shift modulator [17]. This design included a binary phase-shift modulator to achieve a 180° phase shift and a reflection amplifier. However, it still did not realize traditional tag functionality. At an input frequency of 915 MHz and input power of −50 dBm, the reflection gain was 10.2 dB, with a power-added efficiency of just 0.0291%. While the system realizes phase modulation and extends communication range, its reflection gain and power-added efficiency remain quite low, limiting its potential for significant range enhancement.

Although reflection amplifiers have attracted considerable interest, the reported results manifest low gain and limited power-added efficiency. In addition, most studies have focused solely on reflection amplifier design without the implementation of passive sensing tags for long-range communication.

In consideration of all of the above, this paper utilizes discrete components to develop a long-range passive sensing tag that combines multi-source energy harvesting and reflection amplification, effectively enhancing tag sensitivity and reflected signal power, thereby extending the overall communication range of the system.

## 2. Long-Distance Passive Sensing Tag Design

The system architecture of the long-range passive sensing tag, which integrates multi-source energy harvesting and reflection amplification, is illustrated in Figure 1. It comprises an antenna, matching circuit, power splitter, rectifier circuit, energy harvesting and management circuit, demodulation circuit, modulation circuit, microcontroller unit (MCU), hydrogen concentration sensor, and signal conditioning circuit (sensing circuit). Additional components include RF switches, solar cells, operational amplifiers (op amps), and a reflection amplifier. The operational amplifier is applied to amplify the detection signal, hence enhancing the tag’s demodulation sensitivity. The RF switch enables selection between the reflection amplifier and traditional RFID functionality. The reflection amplifier operates under MCU control, with the bias voltage set to a high level when increased reflected power is required. A diode in series connects the output of the solar cell in parallel with the output of the rectifier circuit to the energy harvesting circuit BQ, enabling multi-source energy harvesting. This design not only reduces the tag’s production cost but also remarkably improves its energy harvesting rate when multiple energy sources are available, compared to reliance on a single energy source [22].

The sensing circuit is used to realize the function of information sensing. This paper uses MCU for signal processing; therefore, the passive tag designed in this paper is suitable for most low-power sensors. The sensor output signal can be an analog signal (MCU built-in ADC), or a digital signal using a bus such as I2C or SPI. The specific sensor circuit designed depends on the actual use of sensors; this paper adopts an electric bridge sensor, the output is an analog signal, the sensor output signal is converted into a digital signal through the ADC of the MCU, and the tag sends the digital signal to the reader so as to achieve the sensing function.

In Figure 1, the power splitter distributes the input power in a specified ratio (1:K) between the demodulation circuit and the rectifier circuit. In applications powered solely by RF energy, K can be set to a larger value, allowing more energy to be allocated for energy harvesting. When the system relies only on solar power, K is set to 1, ensuring that all energy supports the demodulation function, contributing to the improvement in demodulation sensitivity. In the multi-source energy harvesting circuit configuration, K is set to 2, enabling balanced energy distribution for both energy harvesting and long-distance communication. Here, K is set to 2 to accommodate for both indoor short-range and outdoor (solar-powered) long-range applications.

### 2.1. Multi-Source Energy Harvesting Circuit Design

In Figure 2, BQ25570 is adopted for the harvesting and management of solar and radio frequency energy. The solar cell is connected to the input pin of the BQ25570 in parallel with the outputs of the diode and RF-DC, forming an integrated energy harvesting circuit. This configuration enables simultaneous energy harvesting, and when the two energies are present at the same time, the charging time is less than when a single energy is present.

### 2.2. Design of High Sensitivity Demodulation Circuit

As shown in Figure 3, this paper presents the structure of the high-sensitivity ASK demodulation circuit, where a voltage-doubling rectification principle is used to enable signal detection and enhance the output voltage. In addition, an operational amplifier is further utilized to amplify the detected signal, significantly improving the sensitivity of the ASK demodulation.

By adding an op amp, the tag sensitivity can be increased to −45.5 dBm, and the forward link (downlink) communication distance *R* can reach 217 m according to Formula (1).
(1)$$P_{tag}=P_{EIRP}+G_{tag}+L_{pol}+10\log{\left(\frac{\lambda }{4\pi R}\right)}^{2}$$

*P_tag_* indicates the received power of the tag, *P_EIRP_* (effective omnidirectional radiated power) indicates the transmitted power of the reader, while *G_tag_* suggests the gain of the tag antenna; *L_pol_* suggests the polarization loss of the transmitting and receiving antennas, and λ indicates the wavelength of electromagnetic wave; and *R* indicates the distance between the reader and the tag.

### 2.3. Design of High-Gain and High-Energy Reflection Amplifier Circuit

The reflection amplifier features a single input port, serving as both the input and output, similar to the injection lock amplifier. Its primary function is to amplify the input signal power, with gain achieved through negative resistance characteristics. Reflection amplifiers are mainly categorized into two types: the first is the tunnel diode-based reflection amplifier, which utilizes the tunnel effect to generate negative resistance characteristics. Tunnel diodes, typically made from materials like gallium arsenide (GaAs) and gallium antimonide (GaSb), offer advantages such as excellent switching characteristics, high speed, and a high operating frequency. However, they require specialized processing technology, are costly, and offer limited adjustability. The second type is the crystal triode (BJT)-based reflection amplifier, which achieves negative resistance through positive feedback. Although its circuit structure is more complex, it offers greater design flexibility and can be implemented with conventional technology. Consequently, this paper focuses on the triode-based reflection amplifier, as it provides more flexibility for adjustment and is better suited for the chip design of sensing tags.

The transistor works in the negative resistance zone under a certain bias voltage and positive feedback, and the impedance *Z_L_* in the negative resistance zone is related to the bias voltage *V_bias_*, input power *P_in_*, and frequency fin, which can be expressed as follows:(2)$$Z_{L}\left(V_{bias},P_{in},f_{in}\right)=-R_{L}+jX_{L},\;R_{L}>0$$

Convert $Z_{L}$ to impedance $Z_{RA}$ at design time
(3)$$Z_{RA}=-R_{RA}+jX_{RA},\;R_{RA}>0$$

If the input impedance of the tag antenna is
(4)$$Z_{a}=R_{a}+jX_{a},\;R_{a}>0$$

The reflection coefficient of the available reflection amplifier is [23]
(5)$$\Gamma_{RA}=\frac{Z_{RA}-Z_{a}^{*}}{Z_{RA}+Z_{a}}$$

Then, the reflected power gain $S_{11}$ of the reflection amplifier is
(6)$$S_{11}=10\log\left({\left|\Gamma_{RA}\right|}^{2}\right)=20\log\left(\left|\frac{Z_{RA}-Z_{a}^{*}}{Z_{RA}+Z_{a}}\right|\right)=20\log\left(\left|\frac{R_{a}+R_{RA}-j(X_{a}+X_{RA})}{R_{a}-R_{RA}+j(X_{a}+X_{RA})}\right|\right)$$

Because both $R_{a}$ and $R_{RA}$ in Formula (6) are greater than 0, it follows that $\left|\Gamma_{RA}\right|$ is greater than 1, then $S_{11}$ > 0 dB. By selecting an appropriate $Z_{RA}$, a stable reflection coefficient (i.e., reflection gain) can be achieved, with the additional power provided by the DC bias of the reflection amplifier, thus satisfying the law of energy conservation. To prevent free oscillation in the reflection amplifier, when choosing $Z_{RA}$, it is necessary to ensure that $R_{a}-R_{RA}>0$ and $X_{a}+X_{RA}\approx 0$ to achieve stable reflection gain. Due to the influence of bias voltage, input power, and frequency, the gain $S_{11}$ is expected to vary with input power.

With the introduction of the reflection amplifier, $\Gamma_{RA}$ serves as the tag reflection coefficient, and the tag modulation loss factor can be obtained from Equation (7):(7)$$\eta =\alpha {\left|\Gamma_{1}-\Gamma_{RA}\right|}^{2}$$

Because when the tag realizes the receiving function, its reflection coefficient $\Gamma_{1}=0$, and Formula (7) is represented by dB as follows:(8)$$10\log(\eta )=10\log(\alpha )+10\log{\left|\Gamma_{1}-\Gamma_{RA}\right|}^{2}=10\log(\alpha )+S_{11}$$

Replace $S_{11}$ with Gamp and bring Equation (8) into Equation (1) to obtain
(9)$$P_{\mathrm{reader}}=P_{\mathrm{EIRP}}+2\left(G_{\mathrm{rag}}+L_{\mathrm{pol}}\right)+G_{\mathrm{amp}}+10\log(\alpha )+G_{\mathrm{reader}}+10\log{\left(\frac{\lambda }{4\pi R}\right)}^{2}$$

Since $G_{amp}>0$, it can be seen that the sensor tag of the integrated reflection amplifier can increase the communication distance of the reverse link.

In this study, the crystal transistor BFU550A is implemented as the reflection amplifier. The circuit configuration of the designed reflection amplifier is shown in Figure 4a, where $V_{cc}$ and $R_{1}$ provide appropriate bias voltages for the transistor. Components $L_{0}$ and $L_{1}$ serve as chokes, while $L_{2}$ and $C_{1}$ enable positive feedback, and $C_{2}$ acts as a coupling capacitor. The ADS (Advanced Design System) was used to optimize the simulation of the reflection amplifier, and the simulation outcomes are illustrated in Figure 4b,c. Specifically, Figure 4b shows the simulation results of $S_{11}$ as it changes with input power at a constant frequency of 923 MHz. It is evident that, for input power levels ranging from −10 dBm to −60 dBm, the relationship between reflection gain and input power remains approximately linear. Figure 4c presents the simulation results of $S_{11}$ in the variation in frequency. The data indicate that when the input power is −60 dBm and the frequency is 923 MHz, $S_{11}$ reaches 45.5 dB in the case where the reflection amplifier circuit is simulated by ADS2016 software. This relationship between gain and input power is described by the following equation:(10)$$G_{amp}=-0.78896P_{in}-1.81809$$

By bringing Equation (1) into Equation (10), the reflection gain at different distances can be obtained:(11)$$G_{amp}=-0.78896\left(P_{EIRP}+G_{tag}+L_{pol}+10\log{\left(\frac{\lambda }{4\pi R}\right)}^{2}\right)-1.81809$$

After taking Formula (11) into Formula (9), Equation (12) is obtained, which represents the tag reflection power received by the collector, under the condition where an integrated reflection amplifier is applied.
(12)$$\begin{array}{c} P_{reader}=0.21104P_{EIRP}+1.21104G_{tag}+1.21104L_{pol}+10\log(\alpha )+\\ G_{reader}+12.1104\log{\left(\frac{\lambda }{4\pi R}\right)}^{2}-1.81809 \end{array}$$

In Equation (12), $P_{reader}$ is the decreasing function of R. Therefore, only the maximum distance that satisfies the collector’s receiving sensitivity needs to be considered. At distances shorter than this maximum, the energy received by the collector exceeds its sensitivity threshold. For instance, when $P_{EIRP}$ = 36 dBm, $G_{tag}$ = 2 dBi, $L_{pol}$ = −3, $G_{\mathrm{reader}}$ = 8 dBi, α = 1/4, and R = 141 m, we find that $P_{\mathrm{reader}}$ ≈ −92 dBm, which meets the requirements for the R420 collector. If R = 302 m, then $P_{\mathrm{reader}}$ = −92 dBm, allowing the R700 collector from Impinj to meet the communication requirements at this distance. However, since the forward link’s communication range is only 217 m—less than the reverse link’s 302 m—the overall system communication distance is limited to 217 m.

## 3. Test and Verification

The framework of the energy harvesting circuit designed for a passive sensing tag integrating multi-source energy harvesting and reflection amplification is shown in Figure 5a, where an antenna is used to receive RF energy and solar cells are utilized to receive light energy. Figure 5b specifies the test results of RF energy and light energy harvesting. The pink curve represents the energy harvesting curve when the RF energy input is −8 dBm; the red curve stands for the energy harvesting curve when light emphasis is 500 Lux; and the black curve describes the energy harvesting curve when RF (−8 dBm) and light energy (500 Lux) are presented at the same time. As illustrated, when RF and light energy exist simultaneously, the harvesting rate of the energy harvesting circuit increases.

The designed reflection amplifier and the passive sensing tag of the integrated reflection amplifier circuit are shown in Figure 6a and Figure 6b, respectively.

As revealed in Figure 7a, the reflection amplifier performance test platform is composed of a reflection amplifier, a DC source, an attenuator, and a vector network analyzer (Agilent N5230A, Palo Alto, Santa Clara, CA, USA). The DC source supplies a DC voltage of 0.828 V to the reflection amplifier, the attenuator adjusts the input power to the reflection amplifier, and the vector network analyzer measures the $S_{11}$ parameter of the reflection amplifier.

In Figure 7b, at an input frequency of $f_{in}$ = 923 MHz, since the minimum output power of the vector network analyzer is −30 dBm, $S_{11}$ stands for the input power of the reflection amplifier measured less than −30 dBm with the help of an attenuator. The data in Figure 7b indicate that $S_{11}$ has an approximately linear relationship with input power (As shown by the blue dashed line in the figure), aligning well with the simulation data. Figure 7c shows the frequency response curve of $S_{11}$ under various input power levels. The curve illustrates that lower input power results in higher output gain of the reflection amplifier. Specifically, when the input power is −60 dBm and the frequency is 919 MHz, $S_{11}$ reaches 44 dB. Based on the current value shown in Figure 7c and a bias voltage of 0.828 V, the DC power $P_{\mathrm{dc}}$ of the reflection amplifier can be calculated. By combining this with the output power $P_{\mathrm{out}}$ of the reflection amplifier, the power-added efficiency (PAE) of the reflection amplifier can be derived as follows [16]:(13)$$PAE=\frac{P_{out}}{P_{dc}}\times 100\%=\frac{P_{in}+G_{amp}}{P_{dc}}\times 100\%$$

The reflection gain testing platform for the long-distance sensing tag is shown in Figure 8a, where an R420 collector, with a receiving sensitivity of $P_{\mathrm{reader}}$ = −84 dBm, is employed. The transmitter end of the collector is connected to a circularly polarized antenna with an 8 dBi gain via an attenuator. The distance between the collector antenna and the tag is maintained at R = 5 m. The attenuator is used to adjust the $P_{EIRP}$ emitted by the collector. The forward and reverse link rates of the system are 50 kbps and 333 kbps, respectively, with M = 4. When the attenuator value $L_{\mathrm{att}}$ is set to −20 dB, the minimum measured power is 2 dBm, allowing for the actual received power at the tag to be calculated as −43.7 dBm. This value is greater than the tag’s demodulation sensitivity of −45.5 dBm, suggesting that the reverse link constrains the system communication distance. Accordingly, from the following formula,
(14)$$G_{amp}=P_{\mathrm{reader}}-P_{\mathrm{EIRP}}-2(G_{\mathrm{reg}}+L_{\mathrm{pol}})-10\log(\alpha )-G_{\mathrm{reader}}-L_{\mathrm{att}}-10\log{\left(\frac{\lambda }{4\pi R}\right)}^{4}$$

The obtained tag reflection gain $G_{\mathrm{amp}}$ ≈ 27.5 dB. Based on this system, if $P_{EIRP}$ = 36 dBm, $G_{\mathrm{amp}}$ ≈ 27.5 dB, $L_{\mathrm{att}}$ = 0, it can be calculated that the communication distance between the long-distance sensing tag and R420 can reach 125 m.

The communication distance test platform between the long-distance sensing tag and R420 is shown in Figure 8b, where $P_{EIRP}$ = 36 dBm, with the longest communication distance measured by the system reaching 96 m. The reason for the actual test communication distance of less than 125 m is that there are more interference factors at longer distances.

Table 1 presents a performance comparison between our designed reflection amplifier and previously reported reflection amplifiers. Apparently, the gain of our reflection amplifier outperforms other designs across various input power levels. In terms of energy efficiency, the proposed reflection amplifier surpasses other transistor-based designs and is only slightly less efficient than the tunnel diode-based approach reported in [16], yet higher reflection gain and longer communication distances are performed.

Table 2 presents a performance comparison between the long-distance sensing tag with an integrated reflection amplifier developed in this paper and similar reported studies. Thanks to the integration of an operational amplifier and a reflection amplifier within the sensing tag, the developed long-distance sensing tag simultaneously enhances sensitivity and reflection power, resulting in a communication range that significantly outperforms that of other similar approaches. Our design of a wireless hydrogen concentration sensing tag featuring a built-in negative resistance reflection amplifier for solar energy harvesting establishes a strong theoretical and experimental foundation for the large-scale deployment of sensing systems based on backscatter communication technology.

## 4. Conclusions

This paper presents the development of a passive long-range sensing tag with the integration of multi-source energy harvesting and reflection amplification. By employing multi-source energy harvesting, the downlink communication range is extended, while reflection amplification enhances the uplink communication distance. It is proved that the tag achieves a reflection gain of 44 dB when receiving a −60 dBm signal at 919 MHz, supporting a communication distance of up to 96 m, which surpasses the range of traditional backscatter communication systems.

## Figures and Tables

**Figure 1 micromachines-16-00018-f001:**
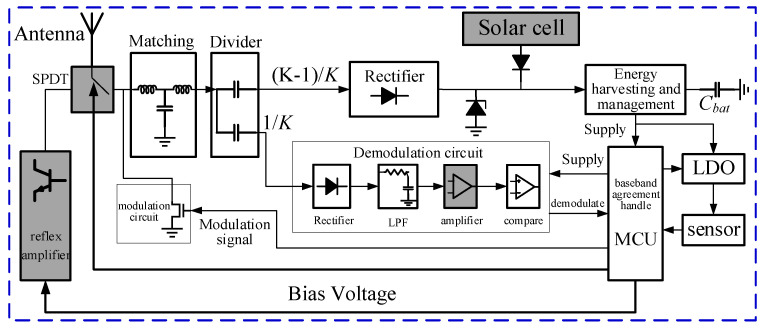
Architecture of a long-range passive sensing tag circuit integrating multi-source energy harvesting and reflection amplification.

**Figure 2 micromachines-16-00018-f002:**
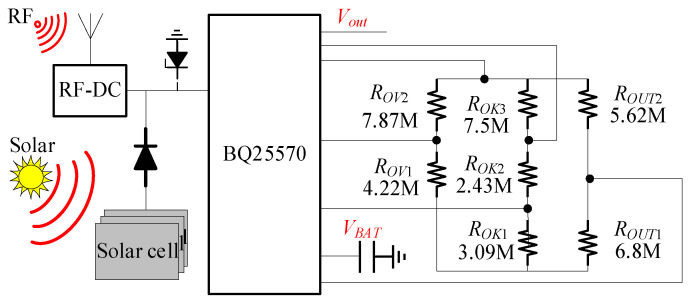
Multi-source energy harvesting circuit design.

**Figure 3 micromachines-16-00018-f003:**
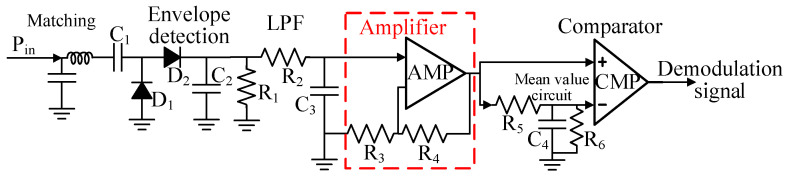
High-sensitivity ASK demodulation circuit.

**Figure 4 micromachines-16-00018-f004:**
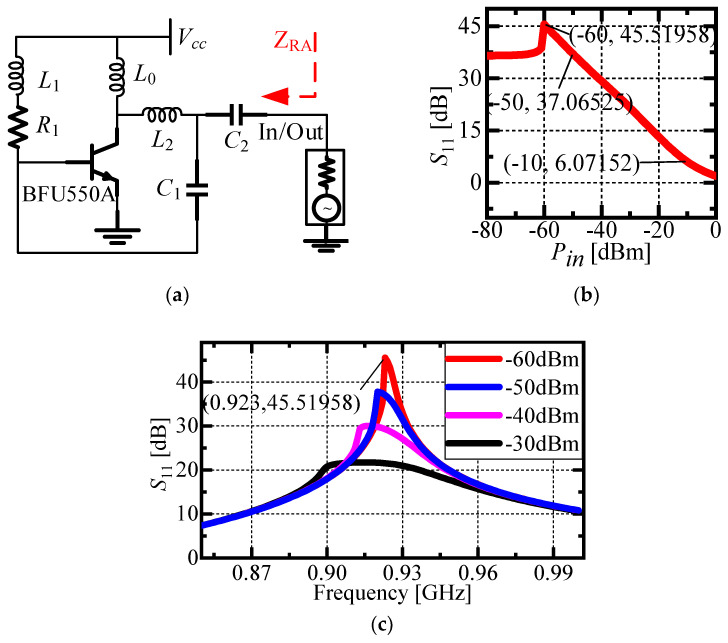
Schematic diagram and performance simulation results of reflection amplifier based on BFU550A. (**a**) Schematic diagram of backscatter amplification circuit; (**b**) curve of gain (dB(S11)) varying with input power at an input signal frequency of 923 MHz; (**c**) the variation curve of gain (dB(S11)) with frequency under different input power.

**Figure 5 micromachines-16-00018-f005:**
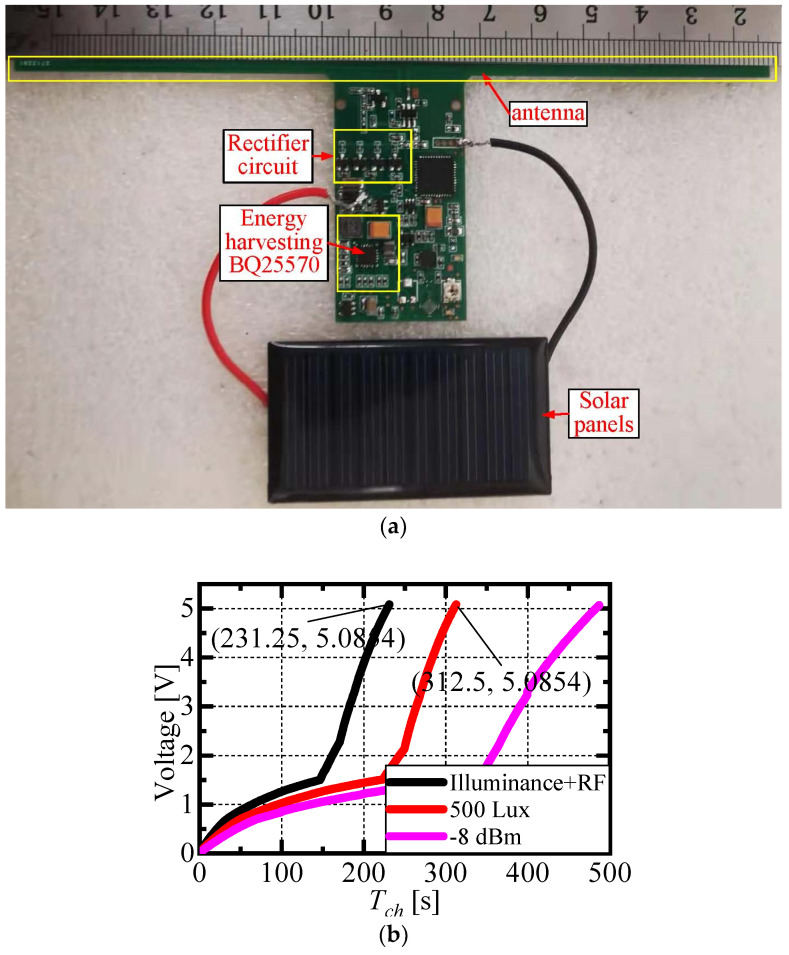
Passive tag with integrated multi-source energy harvesting function. (**a**) Tag energy harvesting circuit module; (**b**) tag energy harvesting performance test results.

**Figure 6 micromachines-16-00018-f006:**
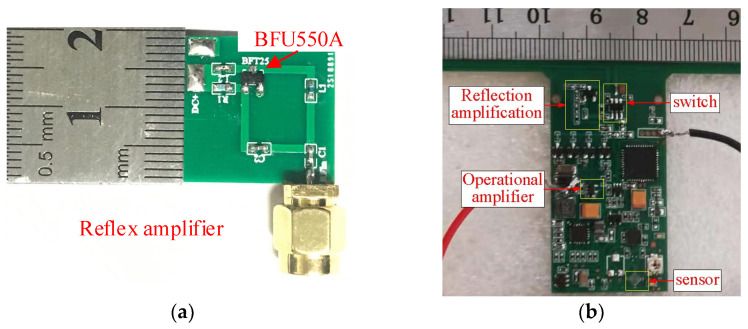
Reflection amplifier and long-distance sensing tag. (**a**) A reflection amplifier object; (**b**) a sensing tag object integrated with a reflection amplifier.

**Figure 7 micromachines-16-00018-f007:**
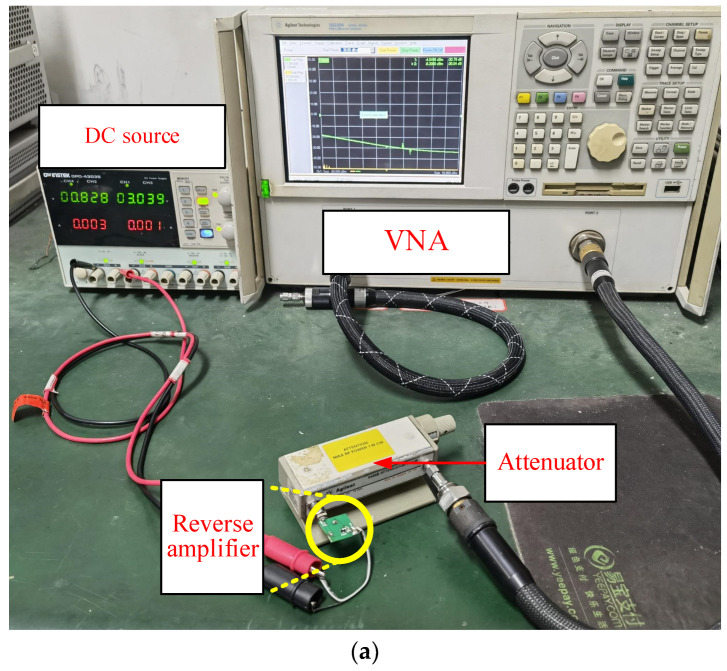
Reflection amplifier performance test platform and test results. (**a**) Reflection amplifier performance test platform; (**b**) curve of reflection gain (dB(S11)) with different input powers ($f_{in}$ = 923 MHz); (**c**) curve of gain (dB(S11)) varying with input frequency under different input powers.

**Figure 8 micromachines-16-00018-f008:**
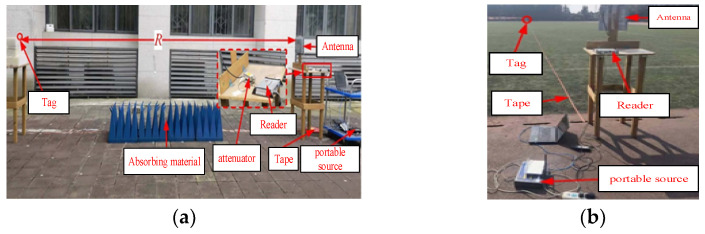
Communication performance test platform between long-distance sensing tag and R420. (**a**) Long-distance sensing tag reflection gain test platform; (**b**) communication distance test platform between long-distance sensing tag and R420.

**Table 1 micromachines-16-00018-t001:** Performance comparison of reverse amplification technology.

Paper	Device	Frequency (GHz)	Input Power (dBm)	Gain (dB)	Output Power (dBm)	Dc Power (mW)	Power-Added Efficiency (PAE)
[14]	BFT25A BJT	0.915	−50	10.2	−39.8	0.325	0.033
[15]	AI201A Tunnel Diode	0.915	−30	13	−17	0.178	11.209
[16]	AI301A Tunnel Diode	0.89	−30	17	−13	0.2	25.055
[17]	BFP840 HBT	5.15	−60	19.1	−40.9	0.5	0.0163
[18]	AI301A GaAs Tunnel Diode	0.8292	−43	20	−23	0.144	3.479
[19]	BFT25A BJT	0.9	−50	29	−21	0.664	1.196
[20]	BJT	0.9	−50	30	−20	0.605	1.653
[21]	MBD5057-E28 Ge Tunnel Diode	5.8	−75	40	−35	0.045	0.702
This work	BFU550A BJT	0.897	−30	23.8	−6.2	1.573	15.248
		0.912	−40	30.3	−9.7	1.524	7.033
		0.918	−50	37	−13	1.515	3.308
		0.919	−60	44	−16	1.515	1.658

**Table 2 micromachines-16-00018-t002:** Comparison of wireless power supply long-distance sensing tag and communication distance of the system.

Paper	Environmental Energy	Frequency (GHz)	Tag Demodulation Sensitivity (dBm)	Collector Sensitivity (dBm)	Communication Distance (m)
[20]	Rf and battery	0.865–0.868	−17.8	−70	15
[7]	Solar energy	0.800~1	−31	−80	20
[8]	Rf and solar	0.868	−35	-	21
[9]	Rf and solar	0.915	<−29	−84	24
[10]	Batteries, RF and solar	0.932	−24.9	−85	25.6
[3]	Rf and solar	0.860~0.960	−25.34	−75	27
This work	Rf and solar	0.920~0.925	−45.5	−84	96

## Data Availability

Data is contained within the article, further inquiries can be directed to the corresponding authors.

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
