# Peer review of "Long-Distance Passive Sensing Tag Design Based on Multi-Source Energy Harvesting and Reflection Amplification"

_micromachines, 2024, doi:10.3390/mi16010018_

Round 1

Reviewer 1 Report

Comments and Suggestions for Authors

1.     The authors focus on the functional realization of the whole system, which leads to the neglect of innovation. Meanwhile, there is nothing new in the system.

2.     The authors should focus on the design of the sensing tag. The authors should introduce multi-energy harvesting in detail.

3.     The manuscript needs to be rewritten, the expression should be more concise and clear, such as: In part 1 introduction, a. line 30-34 the two sentences are a repetition of the same thing, “With the rapid development of ….” “Due to the IoT’s…….”; b. line90-93, “…, illustrated in FIG.12 (a),…. FIG.12 (b)…..” Where is the FIG?; c. line 118-121, “in summary ….. ” this part is introduction. In part 2.1 Multi-source energy collection circuit design, line 151-155, “In this paper,…….” . In part 3, line 257, “Figure 6 (b) presents…..”, line 292, “Table 1 presents…..”, line 300, “Table 2 presents……”

4.     In 2.1 Multi-source energy collection circuit design, where is RF-DC device? Please introduce and highlight in figure 5.

5.     Figure 5 (b), 6 (a), 7 (a) and (b) are very unclear. Therefore, it is not possible to comment on the authenticity of the devices.

6.     As a public research paper, please mention the simulation software, hardware model and test instrument model and manufacturer used in your work.

7.     In Table 1 and 2, the first column should go last.

Comments on the Quality of English Language

3.     The manuscript needs to be rewritten, the expression should be more concise and clear, such as: In part 1 introduction, a. line 30-34 the two sentences are a repetition of the same thing, “With the rapid development of ….” “Due to the IoT’s…….”; b. line90-93, “…, illustrated in FIG.12 (a),…. FIG.12 (b)…..” Where is the FIG?; c. line 118-121, “in summary ….. ” this part is introduction. In part 2.1 Multi-source energy collection circuit design, line 151-155, “In this paper,…….” . In part 3, line 257, “Figure 6 (b) presents…..”, line 292, “Table 1 presents…..”, line 300, “Table 2 presents……”

Author Response

  1. The authors focus on the functional realization of the whole system, which leads to the neglect of innovation. Meanwhile, there is nothing new in the system.

REPLY: Thank you for your suggestions. At present, active wireless sensor networks cover a wide range, but the communication distance of passive sensor systems is extremely limited, and it is difficult to achieve sensor network coverage. Existing studies based on reflection amplification only study the reflection amplification circuit itself, without studying the long sensing technology integrated with emission amplification. Nowadays, there is a lack of systematic reflection to reflect whether the reflection amplification circuit can communicate at any position less than the maximum distance. If the reflection amplification performance is poor, it may lead to the failure of communication at the intermediate distance, which reduces the reliability of the system. At the same time, the lack of practical design and systematic testing can’t provide reference for subsequent work. In this paper, these two problems are realized, and it is theoretically proved that a well-performing reflection amplifier circuit can realize reliable communication at any position in the middle of the longest distance (except for multipath interference), and the tag working distance of integrated reflection amplification is systematically tested, which can provide theoretical and experimental reference data for subsequent research.

  1. The authors should focus on the design of the sensing tag. The authors should introduce multi-energy harvesting in detail.

REPLY: .Thank you for your suggestions. We have added the energy harvesting content, as shown in Figure 1, and added it to Figure 5 of the test verification section of the paper.

The framework of the energy harvesting circuit designed for a passive sensing tag integrating multi-source energy harvesting and reflection amplification is shown in Figure 5 (a), where an antenna is used to receive RF energy and solar cells are utilized to receive light energy. Figure 5(b) specifies the test results of RF energy and light energy harvesting. The pink curve represents the energy harvesting curve when RF energy input is -8 dBm; the red curve stands for the energy harvesting curve when light emphasis is 500 Lux; and the black curve describes the energy harvesting curve when RF (-8 dBm) and light energy (500 Lux) are presented at the same time. As illsurated, when RF and light energy exist simultaneously, the harvesting rate of the energy harvesting circuit increases.

(a)                                                                 (b)

Figure 5 Passive tag with integrated multi-source energy harvesting function. (a) tag energy harvesting circuit module, (b) tag energy harvesting performance test results

  1. The manuscript needs to be rewritten, the expression should be more concise and clear, such as: In part 1 introduction, a. line 30-34 the two sentences are a repetition of the same thing, “With the rapid development of ….” “Due to the IoT’s…….”; b. line90-93, “…, illustrated in FIG.12 (a),…. FIG.12 (b)…..” Where is the FIG?; c. line 118-121, “in summary ….. ” this part is introduction. In part 2.1 Multi-source energy collection circuit design, line 151-155, “In this paper,…….” . In part 3, line 257, “Figure 6 (b) presents…..”, line 292, “Table 1 presents…..”, line 300, “Table 2 presents……”

REPLY:  Thanks for your suggestion, we have revised the full text.

  1. In 2.1 Multi-source energy harvesting circuit design, where is RF-DC device? Please introduce and highlight in figure 5.

REPLY:  Thanks for your suggestion, the RF-DC device has been marked in Figure 5.

  1. Figure 5 (b), 6 (a), 7 (a) and (b) are very unclear. Therefore, it is not possible to comment on the authenticity of the devices.

REPLY:  Thanks to your suggestions, we have optimized figures 5 (b), 6 (a), 7 (a) and (b).

  1. As a public research paper, please mention the simulation software, hardware model and test instrument model and manufacturer used in your work.

REPLY:  Thanks for your suggestion, simulation software and test instruments have been added to the phase position of the paper.

  1. In Table 1 and 2, the first column should go last.

REPLY: Thank you for your suggestion. After careful consideration, we think it is more appropriate to put the literature in the first column. This method of description is also adopted in many papers.

Reviewer 2 Report

Comments and Suggestions for Authors

The presented study is described by the sentence at line 118 in page 2 : “This paper utilizes discrete components to develop a long-range passive sensing tag that combines multi-source energy harvesting and reflection amplification, effectively enhancing tag sensitivity and reflected signal power, thereby extending the overall communication range of the system.”

Each term in equation (1) should be commented. For instance, the radiated power at distance R by an isotropic antenna is uniformly distributed on the surface 4πR². The power flux density reads as P°/ 4πR² where P° is the total radiated power. Consequently the power flux density in dB read as 10 log( (P°/ 4πR²) /Φ°) where Φ° is the reference flux density. 

At line 191 in page 5, “factor (7)” is a typo.

Why is the reflection coefficient in equation (5) defined as (Zʀᴀ – Za*) / (Zʀᴀ + Za)  and not as (Zʀᴀ – Za) / (Zʀᴀ + Za) ? In the latter case, the absolute value of the relection coefficient is greater than 1 only if Rʀᴀ Ra > Xʀᴀ Xa. 

Why is the exponent of R equal to -4 in equation 9 and 14?

Author Response

  1. Each term in equation (1) should be commented. For instance, the radiated power at distance R by an isotropic antenna is uniformly distributed on the surface 4πR². The power flux density reads as P°/ 4πR² where P° is the total radiated power. Consequently the power flux density in dB read as 10 log( (P°/ 4πR²) /Φ°) where Φ° is the reference flux density. 

REPLY:  Thank you for your suggestion, which has been modified in the paper, and the modified content is as follows.

                                        (1)

Where Ptag indicates the received power of the tag, PEIRP (effective omnidirectional radiated power) indicates the transmitted power of the reader, Gtag indicates the gain of the tag antenna, Lpol indicates the polarization loss of the transmitting and receiving antennas, λ indicates the wavelength of electromagnetic wave, and R indicates the distance between the reader and the tag.

  1. At line 191 in page 5, “factor (7)” is a typo.

REPLY:  Thank you for your suggestion, factor(7) should be factor(6), which has been modified in the paper.

  1. Why is the reflection coefficient in equation (5) defined as (Zʀᴀ – Za*) / (Zʀᴀ + Za)  and not as (Zʀᴀ – Za) / (Zʀᴀ + Za) ? In the latter case, the absolute value of the relection coefficient is greater than 1 only if Rʀᴀ Ra > Xʀᴀ Xa. 

REPLY: Thank you for your suggestion. Sometimes the two formulas are the same, but when the formula (Zʀᴀ - Za)/( Zʀᴀ + Za) is used, Za is generally the characteristic impedance and is generally a real number. If Za is a complex number, it is necessary to use (Zʀᴀ - Za*)/( Zʀᴀ + Za). Please refer to the following paper, which has been added in the paper.

[1]K. Kurokawa, "Power Waves and the Scattering Matrix," in IEEE Transactions on Microwave Theory and Techniques, vol. 13, no. 2, pp. 194-202, March 1965

The reflection coefficient is greater than 1 simply because Ra+RRA > ra-RRA, making the numerator greater than the denominator, thus making the reflection coefficient greater than 1.

  1. Why is the exponent of R equal to -4 in equation 9 and 14?

REPLY: Thank you for your suggestion. According to the Friis theorem, the attenuation coefficient of electromagnetic wave in space is inversely proportional to the square of the distance R. In the backscatter communication system, because the distance between the reader and the label is R, the attenuation coefficient of electromagnetic wave from the reader to the label is -2 of R, and the electromagnetic wave will also pass the distance R when it is reflected through the label to the reader. Therefore, the power received by the reader is inversely proportional to the 4th power of the distance R.

Round 2

Reviewer 1 Report

Comments and Suggestions for Authors

1. The authors did not fully answer question 2. Your paper title is "Long distance passive sensing tag design....." The authors should focus on the design of the sensing tag.

2. The pictures in the manuscript are not readable, especially figure 4 (b), figure 5 (a), figure 6 (b) and figure 7 (a), etc.

Author Response

  1. The authors did not fully answer question 2. Your paper title is "Long distance passive sensing tag design....." The authors should focus on the design of the sensing tag.

REPLY: Thank you for your suggestions. We really should increase the sensing design, because the main work of this paper is to design long distance passive tags for sensing functions, after the realization of long distance passive tags, it is more convenient to add sensing functions, so we described in this paper how to achieve the sensing function in the long distance passive tags we designed, as follows:

The sensing circuit is used to realize the function of information sensing. Because, this paper uses MCU for signal processing, therefore, the passive tag designed in this paper is suitable for most low-power sensors. The sensor output signal can be an analog signal (MCU built-in ADC), or a digital signal using a bus such as I2C or SPI. The specific sensor circuit designed depends on the actual use of sensors, this paper adopts an electric bridge sensor, the output is an analog signal, the sensor output signal is converted into a digital signal through the ADC of the MCU, and the tag send the digital signal to the reader, so as to achieve the sensing function.

  1. The pictures in the manuscript are not readable, especially figure 4 (b), figure 5 (a), figure 6 (b) and figure 7 (a), etc.

REPLY: Thank you for your suggestions. We have carefully refined these figures, in which Figures 4(b) and 4(c) have been swapped to make the modified figure 4 look good, thank you.

Reviewer 2 Report

Comments and Suggestions for Authors

the authors addressed my concerns

Author Response

Thank you for your suggestions.